# Serum Long Non-Coding RNAs PVT1, HOTAIR, and NEAT1 as Potential Biomarkers in Egyptian Women with Breast Cancer

**DOI:** 10.3390/biom11020301

**Published:** 2021-02-18

**Authors:** Amal Ahmed Abd El-Fattah, Nermin Abdel Hamid Sadik, Olfat Gamil Shaker, Amal Mohamed Kamal, Nancy Nabil Shahin

**Affiliations:** 1Biochemistry Department, Faculty of Pharmacy, Cairo University, Kasr El-Einy Street, Cairo 11562, Egypt; amal.abdelfattah@pharma.cu.edu.eg (A.A.A.E.-F.); nermin.ibrahim@pharma.cu.edu.eg (N.A.H.S.); nancy.shahin@pharma.cu.edu.eg (N.N.S.); 2Medical Biochemistry and Molecular Biology Department, Faculty of Medicine, Cairo University, Cairo 11562, Egypt; olfat.shaker@kasralainy.edu.eg

**Keywords:** breast cancer, fibroadenoma, PVT1, HOTAIR, MALAT1, NEAT1, PAI-1, OPN

## Abstract

Long non-coding RNAs play an important role in tumor growth, angiogenesis, and metastasis in several types of cancer. However, the clinical significance of using lncRNAs as biomarkers for breast cancer diagnosis and prognosis is still poorly investigated. In this study, we analyzed the serum expression levels of lncRNAs PVT1, HOTAIR, NEAT1, and MALAT1, and their associated proteins, PAI-1, and OPN, in breast cancer patients compared to fibroadenoma patients and healthy subjects. Using quantitative real-time PCR (qRT-PCR), we compared the serum expression levels of the four circulating lncRNAs in patients with breast cancer (*n* = 50), fibroadenoma (*n* = 25), and healthy controls (*n* = 25). The serum levels of PAI-1 and OPN were measured using ELISA. Receiveroperating-characteristic (ROC) analysis and multivariate logistic regression were used to evaluate the diagnostic value of the selected parameters. The serum levels of HOTAIR, PAI-1, and OPN were significantly higher in breast cancer patients compared to controls and fibroadenoma patients. The serum level of PVT1 was significantly higher in breast cancer patients than in the controls, while that of NEAT1 was significantly lower in breast cancer patients compared to controls and fibroadenoma patients. Both ROC and multivariate logistic regression analyses revealed that PAI-1 has the greatest power in discriminating breast cancer from the control, whereas HOTAIR, PAI-1, and OPN have the greatest power in discriminating breast cancer from fibroadenoma patients. In conclusion, our data suggest that the serum levels of PVT1, HOTAIR, NEAT1, PAI-1, and OPN could serve as promising diagnostic biomarkers for breast cancer.

## 1. Introduction 

Breast cancer is the second-ranked leading cause of cancer among women, impacting 2.1 million women each year. It also ranks fifth for causing cancer-associated deaths in women. In 2018, it was revealed that 627,000 women died from breast cancer, accounting for 15% of all cancer deaths among women [1]. While rates of breast cancer are higher among women in developed countries, rates are increasing in every region globally. In rare cases, breast cancer can also affect men [2]. In Egypt, according to the National Cancer Registry Program report in 2013, breast cancer is the most widespread cancer among women accounting for 32% of all women cancer cases. It has been estimated that by 2050, breast cancer incidence in Egypt will increase three-fold relative to the 2013 incidence [3]. The median age at diagnosis is one decade younger than in European countries and North America. The majority of tumors are relatively advanced when presented and, as a result, breast cancer in Egypt has an unfavorable prognosis with a 29% mortality rate [4,5]. 

About 85% of breast cancers occur in women who have no family history of breast cancer. These occur due to genetic mutations that happen as a result of the aging process and life in general. Breast cancer arises from the interaction between multiple genetic variations and environmental factors [6,7]. 

Fibroadenoma is the most common benign breast tumor that can develop at any age with high incidence in young premenopausal women between the ages 15 and 35 years old. The term fibroadenoma is a combination of two words, “fibroma,” which is a tumor made up of fibrous tissue, and “adenoma,” denoting a tumor of gland tissue [8,9]. With time, the fibroadenoma can grow in size or even shrink and disappear. Most fibroadenomas are not associated with an increase in breast cancer risk, with only 0.12–0.3% of fibroadenomas undergoing malignant transformation [10]. 

Long non-coding RNAs (lncRNAs) are a large class of transcribed RNA molecules longer than 200 nucleotides that do not code proteins [11]. High-throughput sequencing and bioinformatics approaches have contributed to an enormous advancement in the comprehensive discovery of lncRNAs. LncRNAs account for about 68% of all transcribed genes [12,13,14]. LncRNAs have unique features, including short lengths, regulatory functions, and tissue-specific expression, which make them candidate biomarkers for disease diagnosis and prognosis [15]. Since the majority of newly discovered lncRNAs have been characterized from a limited number of cell or tissue types, it is predicted that many more lncRNAs are yet to be recognized, particularly in heterogeneous diseases such as different types of human cancer. 

Although most lncRNAs lack an open reading frame and do not code for proteins, proteomic analysis revealed that they can interact with ribosomes and can be translated into functional peptides [16]. Substantial amount of evidence has revealed that they participate in regulating gene expression at transcriptional and/or post-transcriptional levels by interacting with DNA, proteins, or other RNAs. The regulatory role of lncRNAs in gene expression lies within their functions as either miRNA precursors or miRNA sponge through lncRNA–miRNA interactions [17,18]. Increasing evidence indicates that lncRNAs can influence several biological processes, including inflammation, differentiation, proliferation, apoptosis, invasion, and metastasis. Moreover, several lncRNAs that have been previously reported to be involved in various cancers as PVT1 [19], HOTAIR [20], NEAT1 [21], and MALAT1 [22] may contribute to cancer development and progression. 

Plasmacytoma variant translocation 1 gene (PVT1) is a large intergenic lncRNA (300 kb) located on human chromosome 8q24, 57 kb downstream of c-MYC locus. The overexpression of PVT1 was found to be correlated with several types of cancer such as acute myeloid leukemia, Hodgkin lymphoma, breast cancer, ovarian cancer, and pediatric malignant astrocytomas [19,23,24,25]. PVT1 was reported to enhance cell proliferation and inhibit apoptosis in ovarian and breast cancer cell lines and act as a regulator of chemosensitivity in pancreatic cancer [19,26]. In addition, PVT1 was found to increase the expression of plasminogen activator inhibitor-1 (PAI-1) which, in turn, enhances cell migration, invasion, and angiogenesis [19,27]. 

HOX transcript antisense RNA (HOTAIR) is an intergenic lncRNA expressed by the human HOXC locus on chromosome 12q13.13. HOTAIR expression positively correlates with malignant transformation and a poor outcome of several types of cancer as colorectal cancer, breast cancer, ovarian cancer, pancreatic cancer, and gastrointestinal tumors [20,28,29,30]. It was reported that the expression of HOTAIR in cancer cells can be enhanced by osteopontin [31]. Osteopontin (OPN) and phosphoglycoprotein play important roles in tumor growth, invasion, and metastasis [32]. The expression of OPN was reported to be regulated by two important lncRNAs, metastasis-associated lung adenocarcinoma transcript 1 (MALAT1, also known as nuclear-enriched abundant transcript 2, NEAT2) and nuclear enriched abundant transcript 1 (NEAT1) [33,34]. Aberrant expression of MALAT1 and NEAT1 were reported in several types of cancer as hepatocellular carcinoma, pancreatic cancer, ovarian cancer, and colorectal cancer [21,35,36,37]. The precise implication of PVT1/PAI-1 and MALAT1, NEAT1/OPN/HOTAIR networks in breast cancer development is still unclear. 

Substantial support for breast cancer awareness and research has helped to create advances in the diagnosis and treatment of breast cancer. Early detection is critical in order to improve breast cancer outcomes and survival. Therefore, the present study aimed to evaluate the diagnostic potential of serum PVT1, HOTAIR, NEAT1, MALAT1, PAI-1, and OPN in Egyptian women with breast cancer, and also to identify the relationship between the clinicopathological features of breast cancer and the expression of these lnRNAs and associated proteins. 

## 2. Subjects and Methods 

### 2.1. Subjects 

We performed our study on 50 female breast cancer patients with an average age of 53.30 ± 1.03 years (mean ± SD). All patients were recruited from Kasr El-Einy HospitalCairo University, the Department of General Surgery. The diagnosis of all patients was pathologically confirmed after positive mammography. The blood samples were collected before chemotherapy during the period from November 2017 to October 2018. Tumor characteristics, including tumor size, type, grade, tumor-node-metastasis (TNM) stage, and estrogen/progesterone status, were recorded [38,39]. Patients who had received chemotherapy/radiotherapy or who had an acute infection, as well as patients who had cancer at any other site at the time of the selection were excluded from the study. Chest radiography, liver ultrasound, and bone scanning were used to exclude patients with metastatic cancer. 

A total of 25 female fibroadenoma patients were also enrolled from Kasr El-Einy Hospital-Cairo University, the Department of General Surgery, with an average age of 34.16 ± 3.008 years (mean ± SD), who were diagnosed by a mammogram and breast ultrasound. Fibroadenomas were solid, smooth, and painless masses, while aspiration cytology confirmed that they were benign. Additionally, 25 apparently healthy controls, with an average age of 35.32 ± 2.39 years (mean ± SD), were enrolled from the outpatient clinic at Kasr El-Einy Hospital. None of these individuals had been previously diagnosed with malignancies, hypertension, diabetes, or any other diseases. Kasr El-Einy Hospital is a major referral center for all cancer patients. It has cases from all provinces of Egypt. It has the biggest data records of breast cancer cases in the country and can be representative of all of Egypt. 

All subjects enrolled in the study provided informed consent for participation. Also, the study protocol was approved by the Faculty of Pharmacy, Cairo University Ethics Committee (permit number: BC 2057) and conformed to the 1975 Helsinki declaration ethical guidelines. 

### 2.2. Blood Sample Handling 

With the aid of trained laboratory technicians, we collected ~10 mL venous blood samples from each participant and performed a complete blood picture. Two consecutive centrifugation steps were conducted to remove cellular debris (15,000× *g* for 10 min at 4 °C and 20,000× *g* for 3 min at 4 °C). Sera were stored at −80 °C until use for RNA extraction and ELISA assessment of PAI-1 and OPN protein levels. 

### 2.3. Serum lncRNA Assay 

#### 2.3.1. LncRNA Extraction 

For RNA extraction, a volume of 100 µL serum was used as a total sample volume and the extraction was conducted using an miRNeasy extraction kit (Qiagen, Valencia, CA, USA). First, we added 500 µL QIAzol lysis reagent and incubated the whole reaction mixture for 5 min at room temperature. Then, we added 100 µL chloroform, vortexed the mixture for 15 s, and incubated it for 2–3 min at room temperature. Thereafter, we performed centrifugation at 12,000× *g* for 15 min at 4 °C. The upper aqueous phase was then removed, and 1.5 times its volume of 100% ethanol was added. Then, we placed each 700 µL of this mixture in an RNeasy Mini spin column in a 2 mL collection tube and centrifuged at 8000× *g* for 15 s at room temperature. After the mixture had completely passed the column, we added 700 µL of buffer RW1 to each column, and again centrifuged at 8000× *g* for 15 s at room temperature. Next, we added 500 µL buffer RPE to the column and centrifuged it at 8000× *g* for 15 s at room temperature. Afterward, we added another 500 µL buffer RPE to the column and centrifuged it at 8000× *g* for 2 min at room temperature. Subsequently, we placed the column in a new 1.5 mL collection tube and centrifuged it at 8000× *g* for 2 min. Finally, we transferred the column to a new 1.5 mL collection tube and pipetted 50 µL RNase-free water directly onto the column followed by centrifugation for 1 min at 8000× *g* to elute the RNA. After extraction, the sample was treated with DNase to remove any source of DNA before reverse transcription of the RNA into cDNA using DNase Max Kit (Qiagen, Valencia, CA, USA). We then assessed the RNA quantity using a NanoDrop 2000 spectrophotometer (Thermo Scientific, Waltham, MA, USA) at 260/280 nm. 

#### 2.3.2. Reverse Transcription 

Reverse transcription (RT) was performed in a 10-μL final reaction volume with 1 μg RNA (incubated for 60 min at 37 °C, 5 min at 95 °C, and then maintained at 4 °C) using a high-capacity cDNA reverse transcription kit (Applied Biosystems, Foster City, CA, USA) according to the manufacturer’s instructions. 

#### 2.3.3. Quantitative Real-Time Polymerase Chain Reaction 

Quantitative real-time polymerase chain reaction (qRT-PCR) was used to evaluate the expression levels of the serum lncRNAs PVT1, HOTAIR, NEAT1, and MALAT1. The housekeeping glyceraldehyde-3-phosphate dehydrogenase (GAPDH) was used as the endogenous control. The primer sequences used for each of the studied genes are shown in Table 1. For real-time PCR, 3 µL of RT products (cDNA template) was mixed with SYBR Green Master Mix (Qiagen, Valencia, CA, USA) in a final volume of 20 µL with an average primer concentration of 300 nM. An Applied Biosystems 7500 Real Time PCR system (Foster City, CA, USA) was used to perform real-time PCR with the following conditions: 95 °C for 15 min, followed by 40 cycles at 94 °C for 15 s, 55 °C for 30 s, and 70 °C for 34 s. The cycle threshold (CT) is defined as the number of cycles required for the fluorescent signal to cross the threshold in real-time PCR. 

The expression of lncRNAs was computed as the ΔCt value, calculated by subtracting the CT values of GAPDH from the CT values of the target lncRNAs. As there is an inverse correlation between ΔCt and lncRNA expression level, higher ΔCt values were accompanied by decreased lncRNA. The 2^–ΔΔCt^ method of Livak and Schmittgen [34] was used to determine the relative quantitative levels of individual lncRNAs, where ΔΔCt = ∆Ct (sample) − ∆Ct (control average). 

### 2.4. Determination of Serum PAI-1 and OPN Levels by ELISA 

Serum PAI-1 and OPN levels were measured using enzyme-linked immunosorbent assay (ELISA) kits (NOVA, Beijing, China). The serum levels of PAI-1 and OPN were calculated by interpolation from reference curves generated using provided reference standards of known concentrations. All assays were performed in duplicate and complied with the manufacturer’s instructions. The serum levels of PAI-1 and OPN were expressed as ng/mL. 

### 2.5. Statistical Analysis 

All measured parameters were subjected to normality testing using Shapiro–Wilk normality test. Values were expressed as mean ± standard deviation (SD), median (25%–75% percentiles), or number (percentage). A comparison of numerical variables was performed using Student’s *t*-test to compare between two groups and ANOVA with Sidak’s multiple comparison test to compare between the three study groups in case of normally distributed data. The Mann–Whitney U-test or Kruskal-Wallis test followed by Dunn’s multiple comparison were used to compare between two or three groups, respectively, for non-normally distributed data. To compare categorical data, a Chi-squared (X2) test, or Fischer exact test was performed. Multivariate stepwise logistic regression was performed to identify the significant predictors of breast cancer. The receiver operating characteristic (ROC) analysis was used to determine the cut-off values and to analyze the diagnostic utility of the different tested markers. A *p*-value of less than 0.05 was considered statistically significant. All *p*-values are two-sided. Statistical calculations were performed using GraphPad Prism 6.0 (GraphPad Software, CA, USA) and Statistical Package for the Social Science (SPSS, Chicago, IL, USA) software version 15. Power analysis was performed using G*Power software version 3.1.2 (Franz Faul, Kiel University, Germany), employing post hoc Wilcoxon-Mann-Whitney test, with a type I error probability of 0.05. Power analysis is accepted if it is 80%. In addition, we downloaded data from TCGA pan-cancer ATLAS studies in cBioportal, https://www.cbioportal.org/ (Accessed on: 01, February, 2021), for mRNA expression using Hiseq-RNA in breast invasive ductal and lobular carcinoma to compare the value with values obtained from Egyptian breast cancer patients.

## 3. Results 

### 3.1. Demographic and Clinical Features of the Breast Cancer, Fibroadenoma Patients, and Controls 

The demographic profile of the breast cancer, fibroadenoma patients, and the controls is described in Table 2. In brief, the age in breast cancer patients was significantly higher than in the controls (*p* < 0.001) and the fibroadenoma group (*p* < 0.001), whereas no significant difference was observed between the fibroadenoma group and the controls. In addition, a significant difference was observed between breast cancer and fibroadenoma patients in the menopausal status (*p* < 0.0001), family history (*p* < 0.0001), diabetes (*p* < 0.0001), and hypertension (*p* < 0.0001). 

### 3.2. Serum Expression Levels of lncRNAs PVT1, HOTAIR, NEAT1, and MALAT1 

The serum levels of HOTAIR and PVT1 were significantly higher in the breast cancer patients than in the controls at *p* < 0.05, while the serum level of NEAT1 was significantly lower in the breast cancer patients compared to the control group at *p* < 0.01 and no significant difference was observed in the serum level of MALATl in breast cancer patients. 

Comparing breast cancer with fibroadenoma patients, HOTAIR expression levels were significantly higher, whereas NEAT1 expression levels were significantly lower in the breast cancer patients than in the fibroadenoma patients at *p* < 0.0001. No significant difference was observed in the serum levels of PVT1 and MALAT1 of breast cancer patients compared to fibroadenoma. 

Fibroadenoma patients showed significantly lower serum levels of HOTAIR and MALAT1 (*p* < 0.0001), but significantly higher serum levels of PVT1 and NEAT1 (*p* < 0.05) than in the control group as shown in Table 3 and Appendix A.

It is worthy to note that within the breast cancer group, there was a significant increase in the serum expression levels of MALAT1 and HOTAIR in patients with diabetes mellitus, tumor size > 5 and tumor stages three and four compared to the non-diabetic patients, patients with tumor size < 5 and patients with tumor stage two, respectively, at *p* < 0.05 as shown in Table 4. 

We compared our results with the mRNA expression from illumine Hiseq-RNA for the four lncRNAs found in the TCGA database using cBioportal and found a similar increase in the expression level of PVT1 in breast cancer. In contrast, the expression levels of HOTAIR and NEAT1 were not significantly altered in breast cancer patients.

To compensate for the age difference between the breast cancer, control, and fibroadenoma groups, breast cancer patients were stratified by age into patients above and below 50 years old. Then, the breast cancer subgroup below 50 years old was further compared with the control and fibroadenoma groups, where the age difference in this case was non-significant. Surprisingly, such comparison revealed a lack of significant difference in the PVT1 and HOTAIR levels, and, on the contrary, a significantly different MALAT1 level in breast cancer patients compared to the control as shown in Appendix A. We also compared the expression levels between the two breast cancer age subgroups and found them statistically non-significant.

### 3.3. Serum PAI-1 and OPN Levels 

The serum PAI-1 level was significantly higher in the breast cancer patients than in the controls by almost five-fold (*p* = 0.0001). Also, the serum OPN level was 94% higher in the breast cancer patients than in the controls (*p* < 0.05). Both PAI-1 and OPN serum levels were significantly higher in the breast cancer patients compared with the fibroadenoma patients (*p* < 0.05), as manifested by 65% and 75% increments, respectively (Table 5). As shown in Table 4, the statistical analysis between the serum levels of each of PAI-1 and OPN and clinicopathological characteristics revealed a significant association of OPN with age and estrogen/progesterone receptors. A significant elevation of the serum OPN level was observed in patients with age > 50 years and in patients with negative estrogen/progesterone receptor expression. 

Notably, the two breast cancer subgroups stratified by age showed significantly different serum OPN levels, while comparing the younger breast cancer subgroup with the control and fibroadenoma groups revealed no significant difference regarding their serum OPN levels as shown in Appendix A.

### 3.4. Correlation Analysis between the Serum Levels of PVT1, HOTAIR, NEAT1, MALAT1, PAI-1, and OPN in the Breast Cancer Patients 

Serum HOTAIR level was found to be positively correlated with NEAT1 and MALAT1 levels in the breast cancer patients (r = 0.35, *p* < 0.05 and r = 0.51, *p* < 0.001, respectively). Additionally, the serum PAI-1 level was observed to be negatively correlated with HOTAIR and MALAT1 level (r = –0.29, *p* < 0.05 and r = –0.31, *p* < 0.05, respectively) as shown in Table 6. 

### 3.5. Evaluation of the Diagnostic Accuracy PVT1, HOTAIR, NEAT1, MALAT1, PAI-1, and OPN Serum Levels 

The significance of serum PVT1, HOTAIR, NEAT1, MALAT1, PAI-1, and OPN levels as potential diagnostic biomarkers for breast cancer was assessed using a ROC curve. ROC curve analysis showed that PVT1, NEAT1, HOTAIR, PAI-1, and OPN can significantly differentiate between breast cancer and controls, showing an area under the curve (AUC) of 0.67 (95% CI 0.55 to 0.79, cut off > 1.2, *p* < 0.05) for PVT1, an AUC of 0.65 (95% CI 0.53 to 0.77, cut off >1.1, *p* < 0.05) for HOTAIR, an AUC of 0.83 (95% CI 0.73 to 0.93, cut off < 0.87, *p* < 0.001) for NEAT1, an AUC of 0.98 (95% CI 0.94 to 1.01, cut off > 10.8, *p* < 0.0001) for PAI-1, and an AUC of 0.86 (95% CI 0.78 to 0.94, cut off > 30.71, *p* < 0.0001) for OPN. The optimal sensitivity and specificity were 62% and 64% for PVT1, 62% and 64% for HOTAIR, 82% and 80% for NEAT1, 92% and 96 for PAI-1, and 78% and 76% for OPN, respectively (Figure 1).

Our results also revealed that HOTAIR, NEAT1, PAI-1, and OPN can discriminate between breast cancer patients and fibroadenoma patients, showing an AUC of 0.77 (95% CI 0.66 to 0.88, cut off > 15.1, *p* < 0.001), an AUC of 0.73 (95% CI 0.59 to 0.86, cut off < 0.27, *p* < 0.05), an AUC of 0.71 (95% CI 0.59 to 0.82, cut off > 15.10, *p* < 0.05), and an AUC of 0.83 (95% CI 0.73 to 0.92, cut off > 30.13, *p* < 0.0001), respectively. The optimal sensitivity and specificity were 76% and 76% for HOTAIR, 80% and 80% for NEAT1, 64% and 68% for PAI-1, and 80% and 76% for OPN, respectively as shown in Figure 1.

### 3.6. Logistic Regression Analysis 

Univariate and multivariate logistic regression analyses were carried out to assess the predictive values of PVT1, MALAT1, NEAT1, HOTAIR, PAI-1, and OPN in breast cancer (Table 7). In the univariate analysis, serum levels of PVT1, HOTAIR, PAI-1, and OPN were found to be significant predictors associated with breast cancer risk. In the multivariate analysis, PAI-1 emerged as a multivariate predictor of breast cancer risk. Also, the multivariate regression analysis demonstrated that the predictive ability of breast cancer was increased with an AUC of 0.99 (95% CI 0.97 to 1.01, *p* < 0.001). 

Univariate and multivariate logistic regression analyses were also conducted to predict whether fibroadenoma patients will develop breast cancer (Table 8). Serum levels of HOTAIR, PAI-1, and OPN were shown to be significant predictor variables of fibroadenoma progression to breast cancer in both the univariate analysis and the multivariate analysis. In addition, multivariate regression analysis demonstrated that the predictive ability of breast cancer was increased with an AUC of 0.95 (95% CI 0.902 to 0.994, *p* < 0.001).

### 3.7. Power Analysis 

Power analysis was carried out to assess the diagnostic value of PVT1, HOTAIR, NEAT1, MALAT1, PAI-1, and OPN in breast cancer. Serum levels of PVT1, HOTAIR, PAI-1, and OPN were found to have diagnostic powers of 98%, 90%, 80%, and 88% in discriminating breast cancer from the healthy control. In addition, the serum levels of HOTAIR, MALAT1, PAI-1, and OPN were also reported to have a diagnostic power of 98%, 72%, 80%, and 86% in discriminating breast cancer from fibroadenoma patients. 

## 4. Discussion 

Although conventional diagnostic methods such as mammography have resulted in breast cancer reduction by enabling early diagnosis, many patients are diagnosed with an advanced stage and have a poor prognosis. Thus, the search for more sensitive tumor markers is highly demanded to improve screening, diagnosis, and prognostic evaluation of breast cancer. A large number of genetic and epigenetic changes occur during cancer development and progression, including single nucleotide polymorphism (SNP), DNA methylation, and non-coding RNA alterations that can serve as the basis of biomarker discovery. Non-coding RNAs are usually classified according to their size into miRNAs, which are approximately 22 nucleotides long, and lncRNAs, which are more than 200 nucleotides long [40,41,42]. 

In the present study, the serum expression of PVT1 was significantly increased in the breast cancer patients compared to the control subjects. This is in agreement with previous studies reporting significant expression of PVT-1 in gastric cancer and cervical cancer [43,44]. We also found that PVT1 could significantly discriminate between breast cancer patients and control subjects, with a sensitivity of 62%, a specificity of 64%, and an AUC of 0.67. However, PVT1 was not significantly associated with other clinicopathological features and could not discriminate between breast cancer and fibroadenoma patients. 

Several studies implicated PVT1 in aspects of cancer pathophysiology including the observations that rearrangement of the 8q24 region encoding MYC and PVT1 is frequently involved in human prostate, ovarian, and breast cancer [19,45]. Our findings highlight a notable link between serum PVT1 overexpression and breast cancer, lending support to the formerly reported correlation of MYC and PVT1 co-amplification with rapid breast cancer progression and poor clinical survival in postmenopausal women with HER2positive breast cancer [46]. In addition, PVT1 was shown to act independently of MYC, when amplified and overexpressed, through increasing cell proliferation and inhibiting apoptosis in breast and ovarian cancer cell lines [19]. Furthermore, PVT1 was previously found to increase the expression of PAI-1 that enhances cell migration, invasion, and angiogenesis [47], although we could not find any correlation between PVT-1 and PAI-1 in the present study. 

Plasminogen activator inhibitor-1 (PAI-1) is a serine protease inhibitor that regulates the plasminogen activation system by inhibiting tissue-type plasminogen activator and urokinase-type plasminogen activator. Under normal conditions, PAI-1 is mainly produced from platelets and endothelial cells, while in cancer it is produced by tumor cells and nonmalignant cells, including endothelial cells, macrophage cells, or adipocytes in the tumor microenvironment. PAI-1 production can augment tumor growth through inhibition of apoptosis and stimulation of angiogenesis [26]. We observed that the serum level of PAI-1 was significantly increased in the breast cancer patients compared to the fibroadenoma patients and the control subjects. This result is in agreement with Yıldırım et al. 2017 who reported that elevated serum level of PAI-1 is an independent biomarker of endometrial cancer in a Turkish population [48]. Moreover, an elevated preoperative serum PAI-1 level observed in patients with colorectal carcinoma was reported to be significantly associated with a poorer prognosis [49]. In addition, an increase in serum PAI-1 level was significantly associated with histological grade in intracerebral glioma [50]. 

Our results revealed that serum PAI-1 could significantly discriminate between breast cancer patients and control subjects with a sensitivity of 92%, a specificity of 96%, and an AUC of 0.98. In addition, PAI-1 could discriminate between breast cancer and fibroadenoma patients with a sensitivity of 64%, a specificity of 68%, and an AUC of 0.71. In contrast to the formerly reported significant association of PAI-1 with tumor size, pathohistological type, and receptor status of breast cancer [51], our results did not show any significant association between PAI-1 and other clinicopathological features. Our finding regarding the lack of association of PAI-1 level with breast cancer prognosis is also inconsistent with numerous studies showing that high levels of PAI-1 in primary tumor tissue negatively affect the outcome of breast cancer. Zemzoum and co-workers demonstrated the value of tissue PAI-1 as an independent prognostic factor for disease-free survival in node-negative breast cancer [52]. Another study pointed out the association of PAI-1 protein expression in breast cancer tissue with aggressiveness and prognosis for lymph node-negative patients who did not receive chemotherapy [53]. Moreover, Cufer and co-workers depicted the possible utility of PAI-1 level as a biomarker of higher risk of local relapse in breast cancer patients at the time of primary treatment [54]. 

In the present study, we also measured the serum expression levels of HOTAIR, MALAT1, and NEAT1 as possible biomarkers of breast cancer. HOTAIR, MALAT1, and NEAT1 genes are located on chromosomes 12q13.13, 11q13.1, and 11q13.1, respectively. HOTAIR is a well-characterized intergenic lncRNA expressed by the human HOXC locus. It recruits polycomb repressive complex 2 (PRC2) and lysine-specific histone demethylase 1 (LSD1) complexes to chromatin for histone methylation and demethylation processes, respectively, and consequently silencing various tumor suppressor genes [55,56]. We observed that the serum expression level of HOTAIR was significantly higher in the breast cancer patients compared to the fibroadenoma patients and the control subjects. This observation is in agreement with Qiu and co-workers who reported that the increased serum expression of HOTAIR can be used as a prognostic biomarker for ovarian cancer [27]. In addition, it was found that serum HOTAIR could be used as a diagnostic biomarker in gastrointestinal and esophageal squamous cell carcinoma [29,57]. With regard to the clinicopathological features, we observed a significant association between serum HOTAIR expression level and the tumor size. In support of our finding, Zhang and coworkers demonstrated that propofol, a general anesthetic drug, reduced cervical cancer tumor size and promoted cell apoptosis via inhibiting mTOR/p70S6K signaling mediated by HOTAIR [58]. We also demonstrated a significant association between HOTAIR and the tumor stage as has also been reported in colorectal and esophageal squamous cell carcinoma [59,60,61]. The implication of HOTAIR in carcinogenesis could be explained on the basis that the combination of HOTAIR with polycomb repressive complex 2 (PRC2) promoted histone H3K27 methylation of the WIF1 (inhibitor of Wnt signaling pathway) promoter, reducing its expression. As a result, the reduced ß-catenin degradation and the increased T-cell factor/lymphoid enhancer-binding factor levels activate the Wnt/β-catenin signaling pathway. This process enhances tumor cell proliferation, invasion, and metastasis [58]. 

Our data also revealed that serum HOTAIR expression could significantly discriminate between breast cancer patients and control subjects with a sensitivity of 62%, a specificity of 64%, and an AUC of 0.65. In addition, HOTAIR could discriminate between breast cancer and fibroadenoma patients with a sensitivity of 76%, a specificity of 76%, and an AUC of 0.77. This marked diagnostic performance of HOTAIR for breast cancer is in concordance with the formerly reported potential of HOTAIR DNA to detect breast cancer with an AUC of 0.79 [19]. In addition, Sørensen et al. 2013 reported that HOTAIR expression could be used as an independent biomarker for risk assessment of metastasis in estrogen receptor-positive breast cancer patients [62]. Moreover, high expression of HOTAIR reportedly contributed to tamoxifen resistance in breast cancer patients [63]. 

Previous studies have reported the association of the serum expression of NEAT1 and MALAT1 with several types of cancer [21,34,35,64,65,66,67,68]. NEAT1 is a lncRNA that is essential for the formation of nuclear paraspeckles, nuclear bodies enriched in pre-mRNA splicing factors, located in the interchromatin region of mammalian cells [69]. NEAT1 is also required for mammary gland development, lactation, and corpus luteum formation [70,71]. The current study revealed that the serum expression of NEAT1 was significantly lower in the breast cancer patients compared to the fibroadenoma patients and the control subjects. This result is in accordance with Zeng et al. 2014 who reported decreased expression of NEAT1 in acute promyelocytic leukemia [72]. As pointed out by those investigators, the decrease in NEAT1 expression may be explained by the repression of NEAT1 by a transcriptional repressor as promyelocytic leukemia-retinoic acid receptor α (PML-RARα) [73]. In contrast, the serum expression of NEAT1 was significantly increased in colorectal, gastric, prostate, and liver cancer [36,63,72,74]. This discrepancy might be attributed to the dual role of NEAT1 in cancer development [34,75]. Furthermore, our results revealed that NEAT1 could significantly discriminate between the breast cancer patients and control subjects with a sensitivity of 82%, a specificity of 80%, and an AUC of 0.83. In addition, NEAT1 could differentiate between breast cancer and fibroadenoma patients with a sensitivity of 80%, a specificity of 80%, and an AUC of 0.72. However, NEAT1 was not significantly associated with other clinicopathological features. 

MALAT1 is an intergenic lncRNA that was first discovered as a prognostic biomarker for patient survival of stage I lung adenocarcinoma [76]. MALAT1 was also reported to facilitate cell growth, migration, and invasion in several malignancies [77,78,79]. However, our results did not show any significant difference in the serum expression of MALAT1 in the breast cancer patients compared to the fibroadenoma patients and the control subjects. In contrast to our finding, Feng et al. 2016 reported higher expression of MALAT1 in breast cancer tissue compared with adjacent non-cancerous tissue [80]. Also, Huang et al. 2016 found that MALAT1 overexpression was associated with poor survival and tamoxifen resistance in estrogen receptor-positive breast cancer patients [21]. On the other hand, Xu et al. 2015 found lower expression of MALAT1 in breast cancer tissue with an increase in epithelial-mesenchymal transition [65]. As regards the clinicopathological features, we observed a significant association between MALAT1 and comorbidity with diabetes mellitus. This is in agreement with Puthanveetil and co-workers’ report that MALAT1 is involved in hyperglycemia by inducing inflammatory cytokines such as tumor necrosis factor-alpha (TNF-α) [81]. 

In our study, we also measured the serum level of OPN. OPN belongs to a small integrin-binding ligand N-linked glycoprotein family. It is a glycophosphoprotein that plays an important role in the oncogenic and metastatic potential of various cancers [23,24,31]. The expression of osteopontin was found to be upregulated by both MALAT1 and NEAT1 [32,33]. In addition, it was reported that the expression of HOTAIR in cancer cells can be enhanced by OPN [30,32,33]. We observed a significantly higher serum OPN level in the breast cancer patients compared to the fibroadenoma patients and the control subjects. This result is in agreement with a report of increased serum levels of OPN in breast, colon, prostate, and lung cancers [82]. In addition, Rodrigue et al. found that the increased serum level of OPN was associated with a worse prognosis in breast cancer patients [83]. This implication of OPN in breast cancer was further confirmed by our ROC analysis findings which revealed that OPN could significantly discriminate between breast cancer patients and control subjects with a sensitivity of 78%, a specificity of 76%, and an AUC of 0.86. It could also discriminate between breast cancer and fibroadenoma patients with a sensitivity of 80%, a specificity of 76%, and an AUC of 0.83. Regarding the clinicopathological features, we observed a significant association of OPN with age and estrogen/progesterone receptor status. This is in agreement with the previously reported positive interaction between OPN and the estrogen receptor [83]. We also observed a positive correlation between HOTAR and both NEAT1 and MALAT1. In addition, we found a negative correlation between PAI-1 and both HOTAIR and MALAT1. 

In the present study, the multivariate logistic analysis showed PAI-1 as an independent predictor of breast cancer when compared to the control. It also showed HOTAIR, OPN, and PAI-1 as independent predictors of breast cancer when compared to fibroadenoma patients. This result is further supported by our power analysis which revealed a strong diagnostic power for HOTAIR, PAI-1, and OPN, exceeding 80%, to discriminate breast cancer patients from both fibroadenoma patients and controls.

A limitation of our study is the relatively small sample size studied, thereby demanding further confirmatory studies in larger sample sizes. We also urge continued research in this regard for the revelation of more lncRNAs that are pertinent to breast cancer susceptibility in Egyptian patients.

In conclusion, our results revealed that serum PVT1, HOTAIR, NEAT1, PAI-1, and OPN could be promising molecular diagnostic markers for breast cancer. Also, HOTAIR, NEAT1, PAI-1, and OPN could discriminate between breast cancer and fibroadenoma. An important finding of the present study is that PAI-1 is an independent biomarker and has the highest power in discriminating breast cancer patients from controls and fibroadenoma patients. Further studies on more lncRNAs are recommended for more comprehensive screening for genes potentially associated with breast cancer susceptibility.

## Figures and Tables

**Figure 1 biomolecules-11-00301-f001:**
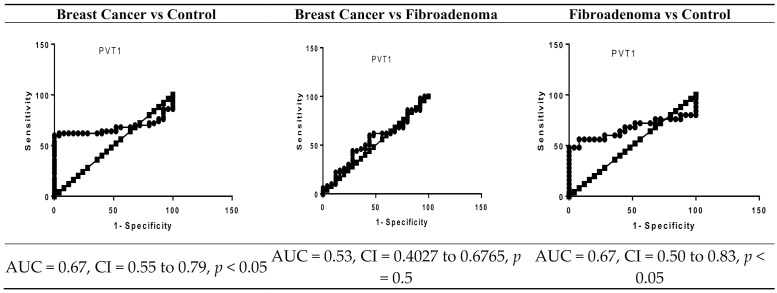
ROC curve analysis of serum PVT1, MALAT1, NEAT1, HOTAIR, PAI-1, and OPN levels in the breast cancer patients (*n* = 50), fibroadenoma (*n* = 25), and healthy controls (*n* = 25). PVT1, plasmacytoma variant translocation 1 gene; HOTAIR, HOX transcript antisense RNA; NEAT1, nuclear enriched abundant transcript 1; MALAT1, metastasis associated lung adenocarcinoma transcript 1; PAI-1, plasminogen activator inhibitor-1; OPN, osteopontin; AUC, area under the curve.

**Table 1 biomolecules-11-00301-t001:** The primer sequences used for qRT-PCR.

Gene Name	Forward Primer	Reverse Primer
*PVT1*	5′-TGAGAACTGTCCTTACGTGACC-3′	5′-AGAGCACCAAGACTGGCTCT-3′
*NEAT1*	5′-TGGCTAGCTCAGGGCTTCAG-3′	5′-TCTCCTTGCCAAGCTTCCTTC-3
*MALAT1*	5′-CTTCCCTAGGGGATTTCAGG-3′	5′-TAGTTGGCATCAAGGCACTG-3′
*HOTAIR*	5′-GGAAAGATCCAAATGGGACCA-3′	5′-CTAGGAATCAGCACGAAGCAAA-3′
*GAPDH*	5′-CTGACTTCAACAGCGACACC-3′	5′-TAGCCAAATTCGTTGTCATACC-3′

PVT1, plasmacytoma variant translocation 1 gene; HOTAIR, HOX transcript antisense RNA; NEAT1, nuclear enriched abundant transcript 1; MALAT1, metastasis associated lung adenocarcinoma transcript 1; GAPDH, glyceraldehyde-3-phosphate dehydrogenase.

**Table 2 biomolecules-11-00301-t002:** Demographic profile of the control, fibroadenoma, and the breast cancer patients.

Parameter		Control *n = 25*	Fibroadenoma *n = 25*	Breast Cancer *n = 50*	*p*-Value
Age		35.32 ± 2.39	34.16 ± 3.00	53.30 ± 1.03	<0.0001 *
Family History	Yes	-	5	28	<0.0001 *
	No	25	20	22	
Menstrual History	Premenopausal	20	20	13	<0.0001 *
	Postmenopausal	5	5	37	
Diabetes Mellitus	Yes	0	0	10	<0.0001 *
	No	25	25	40	
Hypertension	Yes	0	2	10	<0.0001 *
	No	25	23	40	
Tumor Size (cm)	<5	-	-	30	-
	>5	-	-	20	
Tumor Type	Invasive ductal carcinoma	-	-	46	-
	Invasive lobular carcinoma	-	-	4	
Tumor Grade	G I, II	-	-	40	-
	G III	-	-	10	
Tumor-Node-Metastasis (TNM) Classification					
Tumor Stage	T2		-	30	-
	T3, T4	-	-	20	
Lymph Node	N1	-	-	8	-
	N2	-	-	28	-
	N3	-	-	14	-
ER and PR	Positive	-	-	8	-
	Negative	-	-	42	

Data are presented as mean ± standard deviation (SD) or number. Clinical data were analyzed by ANOVA and Chi-squared tests. ***** Indicates statistical significance. *p-*values < 0.05 are considered significant. Tumors were graded according to the modified Bloom–Richardson system; T size of the tumor, T2 (2–5 cm), T3 (>5 cm), T4 (infiltration of the chest wall/skin), N regional lymph node involvement, N1 cancer has spread to one to three axillary lymph node(s), and/or tiny amounts of cancer are found in internal mammary lymph nodes on lymph node biopsy, N2 cancer has spread to four to nine axillary lymph nodes, or cancer has enlarged the internal mammary lymph nodes, N3 cancer has spread to axillary lymph nodes, the internal mammary lymph nodes and/or infraclavicular and supraclavicular lymph nodes; ER and PR, estrogen receptor and progesterone receptor [38,39].

**Table 3 biomolecules-11-00301-t003:** Relative expression levels of serum PVT1, HOTAIR, NEAT1, and MALAT1 in the control, fibroadenoma, and the breast cancer patients.

LncRNA	Control	Fibroadenoma	Breast Cancer	*p-*Value
***PVT1***	0.97 (0.815–1.296)	1.47 ^a*^ (0.108–7.0920)	2.54 ^a*^ (0.731–8.884)	0.03
***HOTAIR***	1.01 (0.815–1.218)	0.31 ^a***^ (0.116–0.553)	1.80 ^a*,b***^ (0.509–3.727)	<0.0001
***NEAT1***	1.01 (0.867–1.1360	1.27 ^a*^ (0.770–1.612)	0.11 ^a***,b***^ (0.0770–0.191)	<0.0001
***MALAT1***	1.01 (0.783–1.266)	0.53 ^a***^ (0.299–0.840)	0.37 (0.109–2.0780)	0.01

The fold change in gene expression was calculated using the 2^–ΔΔCt^ method of Livak and Schmittgen [34] to determine relative quantitative levels of individual lncRNA. Data are expressed as median (25–75% percentiles) and were analyzed by Kruskal-Wallis test followed by Dunn’s multiple comparisons and Mann-Whitney U tests. a. Statistical significance from control group; b. statistical significance from fibroadenoma group; * significance at *p* < 0.05; *** significance at *p* < 0.0001. PVT1, plasmacytoma variant translocation 1 gene; HOTAIR, HOX transcript antisense RNA; NEAT1, nuclear enriched abundant transcript 1; MALAT1, metastasis associated lung adenocarcinoma transcript 1.

**Table 4 biomolecules-11-00301-t004:** Relationship between serum expression levels of PVT1, HOTAIR, NEAT1, MALAT1, PAI-1, and OPN, and the clinicopathological characteristics in breast cancer patients.

Parameter		*PVT1*	*HOTAIR*	*NEAT1*	*MALAT1*	*PAI-1*	*OPN*
Age	<50	4.89	0.45	1.26	0.10	21.94 ± 14.18	35.15 ± 11.68
	>50	2.12	0.31	1.95	0.11	29 ± 18.68	53.49 ± 27.26
	*p*	0.21	0.85	0.58	0.72	0.15	**0.019 ***
Family History	Yes	2.78	0.33	1.64	0.12	25.22 ± 19.39	51.34 ± 26.78
	No	2.34	0.70	2.13	0.11	21.91 ± 9.67	44.55 ± 23.15
	*p*	0.92	0.38	0.39	0.86	0.46	0.35
Diabetes Mellitus	Yes	2.07	0.10	1.35	0.10	27.78 ± 2.12	45.71 ± 22.49
	No	3.27	0.49	1.88	0.11	22.76 ± 14.13	49.02 ± 26.09
	*p*	0.49	**0.01 ***	0.35	0.56	0.37	0.7147
Hypertension	Yes	2.07	0.16	1.87	0.10	20.84 ± 11.28	47.92 ± 21.76
	No	3.27	0.43	1.77	0.11	24.49 ± 16.8	48.46 ± 26.27
	*p*	0.44	0.10	0.78	0.53	0.51	0.95
Liver	Normal	3.31	0.19	0.82	0.08	31.81 ± 21.6	53.12 ± 26.35
	Fatty liver	2.54	0.69	1.88	0.12	22.23 ± 14.29	47.45 ± 25.23
	*p*	0.96	0.08	0.17	0.24	0.11	0.57
Tumor Size	>5	2.91	0.46	2.40	0.10	22.58 ± 10.75	51.33 ± 27.80
	<5	2.27	0.29	1.20	0.15	25.54 ± 2.15	43.89 ± 20.66
	*p*	0.29	0.77	**0.033 ^*^**	0.08	0.52	0.31
Tumor Type	Invasive ductal carcinoma	2.54	0.34	1.80	0.11	22.68 ± 13.58	49.91±25.71
	Invasive lobular carcinoma	4.77	0.59	2.03	0.14	36.29 ± 3.3	30.49 ± 3.598
	*p*	0.92	0.46	0.8204	0.24	0.09	0.14
Tumor Grade	G I, II	2.78	0.36	1.88	0.10	23.55 ± 13.79	46.37 ± 22.6
	G III	2.39	0.42	1.06	0.12	24.63 ± 2.31	56.28 ± 2
	*p*	0.39	0.84	0.79	0.84	0.84	0.27
Tumor Stage	T2	2.90	046	1.20	0.09	22.58 ± 10.75	51.33 ± 27.8
	T3, T4	2.26	0.29	2.40	0.14	25.54 ± 2.15	43.89 ± 20.66
	*P*	0.29	0.77	**0.033 ***	0.08	0.52	0.31
Lymph Node	N1	2.85	0.16	1.12	0.09	22.82 ± 9.733	50.64 ± 17.2
	N2	2.39	0.40	1.842	0.12	25.28 ± 18.67	42.34 ± 23.37
	N3	2.91	0.38	2.231	0.09	21.27±12.46	59.08 ± 13
	*P*	0.88	0.61	0.22	0.32	0.74	0.12
ER and PR	Positive	3.12	0.23	1.26	0.11.	28.73 ± 2.37	30 ± 7.18
	Negative	2.54	0.42	1.88	0.11	22.82 ± 14.02	51.85 ± 25.95
	*P*	0.93	0.46	0.68	0.81	0.33	**0.023 ***

LncRNAs PVT1, HOTAIR, NEAT1, and MALAT1 gene expression are shown as fold change and the data were analyzed by Kruskal-Wallis test or Mann-Whitney U tests and presented as the median, while serum levels of PAI-1 and OPN (ng/mL) were analyzed by ANOVA or Student’s *t*-test and data are presented as or mean ± standard deviation (SD). * Statistical significance at *p* < 0.05; PVT1, plasmacytoma variant translocation 1 gene; HOTAIR, HOX transcript antisense RNA; NEAT1, nuclear enriched abundant transcript 1; MALAT1, metastasis associated lung adenocarcinoma transcript 1; PAI-1, plasminogen activator inhibitor-1; OPN, osteopontin; tumors were graded according to the modified Bloom–Richardson system [39]; T size of the tumor, T2 (2–5 cm), T3 (>5 cm), T4 (infiltration of the chest wall/skin), N regional lymph node involvement, N1 cancer has spread to one to three axillary lymph node(s), and/or tiny amounts of cancer are found in internal mammary lymph nodes on lymph node biopsy, N2 cancer has spread to four to nine axillary lymph nodes, or cancer has enlarged the internal mammary lymph nodes, N3 cancer has spread to axillary lymph nodes, the internal mammary lymph nodes and/or infraclavicular and supraclavicular lymph nodes; ER and PR, estrogen receptor and progesterone receptor [38,39].

**Table 5 biomolecules-11-00301-t005:** Serum levels of PAI-1 and OPN (ng/mL) in the control, the fibroadenoma, and the breast cancer patients.

	Control	Fibroadenoma	Breast Cancer	*p-*Value
PAI-1	3.93 ± 3.37	14.38 ± 3.60 ^a ***^	23.76 ± 15.81 ^a***,b*^	<0.0001
OPN	24.91 ± 6.93	27.64 ± 8.702	48.35 ± 25.23 ^a*,b*^	<0.0001

Data are expressed as mean ± standard deviation (SD) and were analyzed by ANOVA followed by Sidak’s multiple comparison and Student’s *t*-test. ^a^ Statistical significance from control group; ^b^ statistical significance from fibroadenoma group; * significance at *p* < 0.05; *** significance at *p* < 0.0001; PAI-1, plasminogen activator inhibitor-1; OPN, osteopontin.

**Table 6 biomolecules-11-00301-t006:** Correlation analysis between the serum levels of PVT1, HOTAIR, NEAT1, MALAT1, PAI-1, and OPN in the breast cancer patients.

Correlation Analysis	*PVT1*	*HOTAIR*	*NEAT1*	*MALAT1*	*PAI-1*	*OPN*
*PVT1*		r = 0.17 *p* = 0.23	r = 0.27 *p* = 0.05	r = 0.18 *p* = 0.18	r = 0.06 *p* = 0.67	r = 0.19 *p* = 0.18
*HOTAIR*			r = 0.34 *p* = 0.01 *	r = 0.51 *p* = 0.0001 ***	r = −0.29 *p* = 0.04 *	r = 0.013 *p* = 0.92
*NEAT1*				r = 0.27 *p* = 0.05	r = −0.24 *p* = 0.0926	r = −0.12 *p* = 0.38
*MALAT1*					r = −0.30 *p* = 0.02 *	r = 0.14 *p* = 0.32
PAI-1						r = 0.002 *p* = 0.98
OPN						

Data were analyzed by Spearman (for non-parametric data) and Pearson (for parametric data) correlation.* Significance at *p* < 0.05; *** significance at *p* < 0.0001; PVT1, plasmacytoma variant translocation 1 gene; HOTAIR, HOX transcript antisense RNA; NEAT1, nuclear enriched abundant transcript 1; MALAT1, metastasis associated lung adenocarcinoma transcript 1; PAI-1, plasminogen activator inhibitor-1; OPN, osteopontin.

**Table 7 biomolecules-11-00301-t007:** Logistic regression analysis of PVT1, HOTAIR, NEAT1, MALAT1, PAI-1, and OPN to predict breast cancer risk.

Parameter (Control)	Coefficient (β)	SE	Wald (x^2^)	*p*-Value	Odds Ratio	95% CI
**Univariate Analysis**						
*PVT1*	0.62	0.26	5.63	0.018 *	1.86	1.11–3.12
*HOTAIR*	0.58	0.26	5.04	0.02 *	1.79	1.07–3.00
*NEAT1*	0.01	0.12	0.002	0.967	0.99	0.775–1.27
*MALAT1*	0.16	0.12	1.70	0.192	1.17	0.92–1.50
PAI-1	0.63	0.16	14.67	0.0008 **	1.88	1.36–2.59
OPN	0.15	0.04	13.86	0.001 **	1.07	1.07–1.26
**Multivariate Analysis**						
PAI-1	0.44	0.15	7.99	0.01 *	1.56	1.14–2.12

* Significance at *p* < 0.05; ** significance at *p* < 0.001; PVT1, plasmacytoma variant translocation 1 gene; HOTAIR, HOX transcript antisense RNA; NEAT1, nuclear enriched abundant transcript 1; MALAT1, metastasis associated lung adenocarcinoma transcript 1; PAI-1, plasminogen activator inhibitor-1; OPN, osteopontin; SE, standard error.

**Table 8 biomolecules-11-00301-t008:** Logistic regression analysis of PVT1, HOTAIR, NEAT1, MALAT1, PAI-1, and OPN to predict breast cancer risk in patients with fibroadenoma.

Parameter (Fibroadenoma)	Coefficient (β)	SE	Wald (x^2^)	*p*-Value	Odds Ratio	95% CI
**Univariate Analysis**						
*PVT* * 1 *	0.04	0.04	0.70	0.40	1.04	0.94–1.14
*HOTAIR*	1.03	0.35	8.67	0.003 *	2.80	1.41–5.57
*NEAT1*	0.11	0.15	0.54	0.45	1.12	0.82–1.53
*MALAT1*	0.37	0.260	2.09	0.14	1.45	0.87–2.42
PAI-1	0.13	0.04	7.60	0.006 *	1.14	1.03–1.25
OPN	0.08	0.03	7.66	0.006 *	1.09	1.02–1.15
**Multivariate Analysis**						
*HOTAIR*	1.35	0.47	8.19	0.004 *	3.88	1.53–9.81
PAI-1	0.19	0.07	6.49	0.01 *	1.21	1.04–1.40
OPN	0.12	0.05	6.76	0.009 *	1.13	1.03–1.25

* Significance at *p* < 0.05; PVT1, plasmacytoma variant translocation 1 gene; HOTAIR, HOX transcript antisense; RNA; NEAT1, nuclear enriched abundant transcript 1; MALAT1, metastasis associated lung adenocarcinoma transcript 1; PAI-1, plasminogen activator inhibitor-1; OPN, osteopontin; SE, standard error.

## Data Availability

The data presented in this study are available on request from the corresponding author, amal.ibrahim@pharma.cu.edu.eg.

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
