# Peer review of "Serum Long Non-Coding RNAs PVT1, HOTAIR, and NEAT1 as Potential Biomarkers in Egyptian Women with Breast Cancer"

_biomolecules, 2021, doi:10.3390/biom11020301_

Round 1

Reviewer 1 Report

The manuscript presented by Amal Ahmed Abd El-Fattah et al, entitled “Serum long non-coding RNAs PVT1, HOTAIR and NEAT1 as potential biomarkers in Egyptian women with breast cancer” describes the analysis of serum expression levels of lncRNAs PVT1, HOTAIR, NEAT1 and MALAT1, and their associated proteins, PAI-1 and OPN in breast cancer patients compared to fibroadenoma patients and healthy subjects.  The authors suggest that the serum levels of PVT1, HOTAIR, NEAT1, PAI-1, and OPN could serve as promising diagnostic biomarkers for breast cancer.

In general, the material is well organized and easy to be followed and understood.

The results are clear and satisfactory.

The tables and the graphics are informative and very well organized.

Recommendation: It will be nice if the authors compare and validate their data with a larger patient cohort databases such as TCGA.

In my opinion, the current version of the manuscript is suitable for publishing in Biomolecules.

Author Response

Response to reviewers' comments

All changes made are found in yellow highlighter in the revised manuscript: biomolecules-1078836

Reviewer #1:

It will be nice if the authors compare and validate their data with a larger patient cohort databases such as TCGA.

In response to this comment, we compared our lncRNA expression results with the TCGA database in the “Results” section as follows:

“We compared our results with the mRNA expression from illumine Hiseq-RNA for the four lncRNA found in the TCGA database using cBioportal and found a similar increase in the expression level of PVT1 in breast cancer. In contrast, the expression levels of HOTAIR and NEAT1 were not significantly altered in breast cancer patients”.

I also added on the methods “In addition, We downloaded data from TCGA pan cancer ATLAS studies in cBioportal, https://www.cbioportal.org/, for mRNA expression using Hiseq-RNA in breast invasive ductal and lobular carcinoma to compare the value with values obtained from Egyptian breast cancer patients”.

We did not find published data on PAI-1 and OPN in the TCGA, so there data were not included in the comparison.

Reviewer 2 Report

    The manuscript entitles “Serum long non-coding RNAs PVT1, HOTAIR and NEAT1 as potential biomarkers in Egyptian women with breast cancer” by Abdelfattah et al. analyzed the serum expression levels of lncRNAs PVT1, HOTAIR, NEAT1 and MALAT1, and their associated proteins PAI-1 and OPN, in breast cancer patients compared to fibroadenoma patients and healthy subjects. As result, they found that the serum levels of HOTAIR, PAI-1 and OPN were significantly higher, and NEAT1 was significantly lower in breast cancer patients compared to controls and fibroadenoma patients. In contrast, the serum level of PVT1 was significantly higher in breast cancer patients than in the controls.

    The paper showed interesting results; however, some critical points and/or corrections must be performed to paper's acceptance, as reported below:

Major:

Expression Analysis:

    In expression analysis, primers used to PVT1 are specific for cDNA, but NEAT1, MALAT1, HOTAIR and GAPDH primers are not specific, and they can amplify DNA.

    It’s essential to solve this problem. One option is to include DNAse treatment and to analyse amplification without reverse transcription (RT minus). This step proves the absence of DNA contamination.           

    It’s also important to use more than one endogenous control. There is no optimal endogenous, and more than one increases the feasibility of comparative qPCR analysis.     

Control Group:

    The average age of breast cancer patients (53.30 ± 1.03) is different (P<0.001) compared to age of Fibroadenoma (34.16 ± 3.00) and control (35.32 ± 2.39).

    It’s known that age could change RNA expression. Inclusive, in the present manuscript, OPN expression is elevated in patients with age >50 years (P=0.019).

    Based on this, it’s possible that age differences also influenced the differences between BC and control groups. Considering this, it would be interesting to include more individuals in control groups with age-similar with BC patients. An exciting analysis could compare the expression levels of these lncRNAs in control groups with average age ~35 years and ~50 years old.

Minor:

- In Introduction, it is written that “(lncRNAs) are a large class of transcribed RNA molecules longer than 200 nucleotides that do not code proteins”, but it is important to include the information that some lncRNAs may produce functional peptides;

- Included the concentration of RNA included in Reverse Transcription and primer concentration in qPCR;

- The information of table 3 (relative expression) could be more visual in a dot blot graph.

- Better discuss the correlation of OPN, NEAT1, MALAT1 and HOTAIR in the “Discussion” section.

Author Response

Response to reviewers' comments

All changes made are found in yellow highlighter in the revised manuscript: biomolecules-1078836

Reviewer #2:

  1. In expression analysis, primers used to PVT1 are specific for cDNA, but NEAT1, MALAT1, HOTAIR and GAPDH primers are not specific, and they can amplify DNA.

In fact, we extracted the RNA using Qiagen miRNeasy extraction kit and the quality of the yield that was used for further analysis was further checked using the Nanodrop (with average 260/280: 2.4) as recommended by Fleige and Pfaffl [1].

  1. Fleige, S.; Pfaffl, M.W. RNA integrity and the effect on the real-time qRT-PCR performance. Mol. Aspects Med. 2006, 27, 126–139, doi:10.1016/j.mam.2005.12.003.
  2. It’s also important to use more than one endogenous control. There is no optimal endogenous, and more than one increases the feasibility of comparative qPCR analysis.

We thank the reviewer for the insightful comment that we will definitely take into consideration in future studies. We will accordingly take this into account while estimating the required sample volumes and while considering funding issues. However, currently, due to the limited funding, in addition to time limits and the difficulties associated with reagent shipping due to the COVID situation we are facing, the authors will not be able to employ additional endogenous controls at the moment, but will fully take this suggestion in their account in future grant applications.

  1. The average age of breast cancer patients (53.30 ± 1.03) isdifferent (P<0.001) compared to age of Fibroadenoma (34.16 ±3.00) and control (35.32 ± 2.39).]

Due to the current situation, it was difficult to collect more samples from elderly people in addition to funding limits. As a compensation for the age difference, we stratified the breast cancer patients into patients above and below 50 years old, and the subgroup below 50 years old was further compared with the control and fibroadenoma groups as the age difference with this subgroup was non-significant.

This has been clarified in the revised manuscript in the “Results” section as follows:

“To compensate for the age difference between the breast cancer, control and fibroadenoma groups, breast cancer patients were stratified by age into patients above and below 50 years old. Then, the breast cancer subgroup below 50 years old was further compared with the control and fibroadenoma groups, where the age difference in this case was non-significant. Surprisingly, such comparison revealed lack of significant difference in the PVT1 and HOTAIR levels, and, on the contrary, a significantly different MALAT1 level in breast cancer patients compared to the control as shown in Supplementary Table 1. We also compared the expression levels between the two breast cancer age subgroups and found them statistically non-significant.”

Regarding serum OPN level, the following part was added to the “Results” section:

“Notably, the two breast cancer subgroups stratified by age showed significantly different serum OPN levels, while comparing the younger breast cancer subgroup with the control and fibroadenoma groups revealed no significant difference regarding their serum OPN levels as shown in Supplementary Table 2 .

Supplementary Table 1. Relative expression levels of serum PVT1, HOTAIR, NEAT1, and MALAT1 in the control, fibroadenoma and the breast cancer patients.

LncRNA

Control

Fibroadenoma

Breast cancer (Age < 50)

P value

PVT1

0.9686

(0.8145-1.296)

1.468 a*

(0.7967-7.092)

4.737

(0.729-9.169)

0.0785

HOTAIR

1.01

(0.815-1.218)

0.31  a***

(0.116-0.553)

1.202

(0.1452-3.759)

0.0008

NEAT1

1.01

(0.867-1.1360

1.27 a*

(0.770-1.612)

0.1011a***,b***

(0.07536-0.1496)

< 0.0001

MALAT1

1.01

(0.783-1.266)

0.53 a***

(0.299-0.840)

0.2325a*

(0.1584-1.027)

0.0002

The fold change in gene expression was calculated using 2–ΔΔ(Ct) method of Livak and Schmittgen [33] to determine relative quantitative levels of individual lncRNA. 

Data are expressed as median (25%-75% percentiles) and were analyzed by Kruskal-Wallis test followed by Dunn's multiple comparisons and Mann-Whitney U tests.

  1. Statistical significance from control group
  2. Statistical significance from fibroadenoma group

* Significance at P < 0.05

*** Significance at P < 0.0001.

PVT1, Plasmacytoma variant translocation 1 gene; HOTAIR, HOX transcript antisense RNA; NEAT1, Nuclear enriched abundant transcript 1; MALAT1, Metastasis associated lung adenocarcinoma transcript 1.

Supplementary Table 2. Serum levels of PAI-1 and OPN (ng/ml) in the control, the fibroadenoma and the breast cancer patients.

Control

Fibroadenoma

Breast cancer

P value

PAI-1

3.93 ± 3.37

14.38 ± 3.60a***

29.84 ± 22.5a***,b*

<0.0001

OPN

24.91±6.93

27.64±8.702

30.25 ± 6.763

0.1764

Data are expressed as means ± standard deviation (SD) and were analyzed by ANOVA followed by Dunn's multiple comparisons and Student’s t-tests.

  1. Statistical significance from control group
  2. Statistical significance from fibroadenoma group

* Significance at P < 0.05

*** Significance at P < 0.0001

PAI-1, plasminogen activator inhibitor-1; OPN, Osteopontin.

  1. In Introduction, it is written that “(lncRNAs) are a large class oftranscribed RNA molecules longer than 200 nucleotides that do notcode proteins”, but it is important to include the information thatsome lncRNAs may produce functional peptides.

Done. In response to this comment, the following statement has been added to the “Introduction” section:

“Although most of lncRNAs lack open reading frame and do not code for proteins, proteomic analysis revealed that they can interact with ribosomes and can be translated into functional peptide [16].”

  1. Ruiz-Orera, J.; Messeguer, X.; Subirana, J.A.; Alba, M.M. Long non-coding RNAs as a source of new peptides. Elife 2014, 3, 1–24, doi:10.7554/eLife.03523.

  1. Included the concentration of RNA included in ReverseTranscription and primer concentration in qPCR.

Done. In the “Subjects and Methods” section, we clarified that we used 1 μg RNA for reverse transcription, and an average primer concentration of 300 nM. This has been clarified as follows:

“Reverse transcription (RT) was performed in a 10-μl final reaction volume with 1 μg RNA (incubated for 60 min at 37 °C, 5 min at 95 °C, and then maintained at 4 °C)” and “For real-time PCR, 3 µl of RT products (cDNA template) was mixed with SYBR Green Master Mix (Qiagen, Valencia, CA, USA) in a final volume of 20 µl with average primer concentration 300 nM”

  1. The information of table 3 (relative expression) could be morevisual in a dot blot graph.

Yes, we will add the scatter blot graph as supplementary figure as shown below:

Supplementary Fig. 1. Relative expression levels of serum PVT1, HOTAIR, NEAT1, and MALAT1 in the control, fibroadenoma and the breast cancer patients

  1. Better discuss the correlation of OPN, NEAT1, MALAT1 andHOTAIR in the “Discussion” section.

Yes, thank you for recommendation, we added in the discussion “We also observed a positive correlation between HOTAR and both NEAT1 and MALAT1. In addition, we found a negative correlation between PAI-1 and both HOTAIR and MALAT1”.

Reviewer 3 Report

In the manuscript entitled “Serum long non-coding RNAs PVT1, HOTAIR and NEAT1 as potential biomarkers in Egyptian women with breast cancer" the authors suggested that the serum levels of PVT1, HOTAIR, NEAT1, PAI-1 and OPN could serve as promising diagnostic biomarkers for breast cancer. The data revealed that PAI-1 has the greatest power in discriminating breast cancer from the control, whereas HOTAIR, PAI-1 and OPN have the greatest power in discriminating breast cancer from fibroadenoma pa-tients. A limitation of study is the relatively small sample size studied.

However, we suggest some corrections:

  • In the references section, the numbers reported wrong, some references have been divided, altering all the numbering indicated in the text. The authors must double-check and correct the references.

The study is interesting and rich in content, it considered suitable for publication in Biomolecules but after minor revisions.

Author Response

Response to reviewers' comments

All changes made are for the revised manuscript: biomolecules-1078836

Reviewer #3:

In the references section, the numbers reported wrong, somereferences have been divided, altering all the numbering indicatedin the text. The authors must double-check and correct thereferences.

Done. All references have been revised and adjusted and accordingly corrected in the revised manuscript.

Round 2

Reviewer 2 Report

   Most points were discussed and were satisfactory. However, the possibility of genomic DNA amplification by primers and the absence of information about the use of DNAse cast in doubt the results.
   It is essential to answer if DNAse was used during or after RNA extraction to guarantee that qPCR does not amplify DNA and completely changing the results.  

Author Response

Response to reviewers' comments

All changes made are found in yellow highlighter in the revised manuscript: biomolecules-1078836

Reviewer #2:

  • It is essential to answer if DNAse was used during or after RNA extraction to guarantee that qPCR does not amplify DNA and completely changing the results.

After RNA extraction, we used, DNase Max Kit (Qiagen, Valencia, CA, USA) to secure integrity of our extracted RNA, I added this to the method section “After extraction, the sample was treated with DNAse to remove any source of DNA before reverse transcription of the RNA into cDNA using DNase Max Kit (Qiagen, Valencia, CA, USA)”.